# Enhancement of Diversity in Production and Applications Utilizing Electrolytically Polymerized Rubber Sensors with MCF: The Second Report on Various Engineering Applications

**DOI:** 10.3390/s20174674

**Published:** 2020-08-19

**Authors:** Kunio Shimada, Ryo Ikeda, Hiroshige Kikura, Hideharu Takahashi

**Affiliations:** 1Faculty of Symbiotic Systems Sciences, Fukushima University, 1 Kanayagawa, Fukushima 960-1296, Japan; 2Institute of Innovative Research, Tokyo Institute of Technology, 2-12-1 Ookayama, Meguro-ku, Tokyo 152-8550, Japan; ikeda.r.ah@m.titech.ac.jp (R.I.); kikura@lane.iir.titech.ac.jp (H.K.); htakahashi@lane.iir.titech.ac.jp (H.T.)

**Keywords:** sensor, diene rubber, texture, biology, cosmetics, electrolytic polymerization, magnetic compound fluid (MCF)

## Abstract

We investigated the proposed hybrid skin (H-Skin) for the requirement of haptic sensibility in rubber using our proposed consummate fabrication process together with a multi-layered magnetic compound fluid (MCF) rubber and stocking-like porous rubber permeated by liquids, which was demonstrated in our previous report. The objective was to assess its applicability to sensing normal force and temperature, as well as fields dominated by shear force. For normal force, we investigated the piezo-electricity and electric current induced voltage, as well as the piezo-resistivity of the MCF rubber sensor under pressure. Additionally, we clarified the viability of measuring the softness and texture of materials using the MCF rubber sensor. For the shear motion, we clarified the characteristics of the friction coefficient using the MCF rubber sensor. The MCF rubber sensor can capture the reactions of paper, cloth, convex- and concave-shaped objects such as plant leaves and metal, and the skin of the human finger. Therefore, it is useful to investigate its texture and biological surfaces. Our obtained outstanding results indicated the feasibility of sensing the surface texture for any material in fields such as paper, fashion, apparel manufacturing, and cosmetic industries, which was impossible until now.

## 1. Introduction

In our first report, we investigated the viability of the electrolytic polymerization technique for the solidification of diverse rubbers using a combination of various surfactants and water-insoluble magnetic fluids (MFs) via the consummate fabrication of a magnetic compound fluid (MCF) rubber sensor [1]. MCF has nm-ordered magnetite (Fe_3_O_4_) due to MFs and μm-ordered metal particles [2]. MCF rubber is the rubber that incorporates MCF. By the application of a magnetic field, magnetic clusters of Fe_3_O_4_ and metal particles are created in the rubber under solidification from rubber latex [1]. The solidification was presented with several combinations of various rubbers and MFs using various emulsifiers. Rubber can be categorized into diene and non-diene rubbers, or natural and synthetic rubbers. We demonstrated the feasibility of silicone rubber (Q) and urethane rubber (U) as non-diene rubbers, in addition to natural rubber (NR) and chloroprene rubber (CR) as diene rubbers. In general, diene rubber is water-soluble and non-diene is water-insoluble. Some MFs are water-soluble and others are water-insoluble. The various arrangements of water-soluble or water-insoluble rubbers and water-soluble or water-insoluble MFs can be combined with emulsifiers, i.e., polyvinyl alcohol (PVA), whose anionic characteristics are predominantly suitable for excellent combinations and electrolytic polymerization. Moreover, for electrolytic polymerization, by combining either hydrate Na_2_WO_4_·2H_2_O or water, or a mixture of them, we can obtain an MCF rubber with many pores, such as stocking [1]. Rubber stocking can also percolate through liquids or involve any liquids in it. Using these combinations of emulsifiers and permeated rubber stockings, and by introducing a previously proposed adhesion technique of electric wires to the MCF rubber [3], we proposed the fabrication process of an MCF rubber sensor [1]. In this report, as we make the technique and process applicable to diverse fields, we attempt to apply the fabricated MCF rubber sensor to fields related to normal and shear forces, as well as temperature.

Most sensors are used to sense normal force and temperature in various robotic fields and measurement instruments. These sensors also perform the haptic sensing like human skin [4]. Because the human skin has five types of touch sensations—i.e., tactile, baresthesia, algometry, warm, and cold—sensing force and temperature are required. As for the sensing force, most conventional sensors have the same sensibility as normal force. However, sensing shear force is also significant enough to be related to typical examples such as tactile devices, sensing the roughness of human skin or various types of paper (such as toilet paper, newspapers, and paper diapers [5]), cloths [6], and cosmetics [7].

On the one hand, few studies have been conducted on sensing the texture of paper, cloth, and biological skin using haptic sensors. If we can evaluate the texture of such objects, the sensing technique could contribute to enterprising development of their production. On the other hand, measuring surface roughness of an automobile, airplane, ship, etc., a sensor might be wearable and convenient enough to measure and protect the shape of the measured product. Because conventional instrument measures the surface roughness by using a thin diamond needle that scratches the surface of the workpiece, the workpiece specimen must be cut from the object’s surface. Other studies on wearable tactile sensors for surface roughness focus on the tactile sensor, such as piezo-electric polyvinylidene difluoride (PVDF) film sensors [8], pressure conductive sensors, and six-axis acceleration sensors [9]. These sensors are based on the sensitivity of normal force because they utilize the data obtained by calculating multiple normal forces [10]. To measure the shear force, strain gauges, or piezo-electric elements are hierarchized and obtained from normal forces [11,12,13,14]. Because of these elements, the sensor is complicated in structure and vulnerable to extrinsic mechanical forces. Therefore, it lacks extension and is not suitable for applications where the sensor must measure shear motion.

Few novel sensors have been produced using single materials and simpler structures. For example, for the measurement of skin roughness in the beauty industry, a sensor that utilizes strain gauges and a PVDF piezo-polymer film were proposed for the measurement of skin conditions [15], and a sensor fabricated from conductive rubber with carbon powder in Q-type rubber and polyvinyl chloride was used for surface texturing [16]. Therefore, a simple production using a single material, such as only rubber, is required to obtain and evaluate the condition of paper, cloth, skin after applying make-up, or the quality of human hair, as measurable numerical data. In addition, as those objects are vulnerable enough to be soft, the rubber sensor must be sensible to the softness of the objects. In this paper, we address the applicability of the MCF rubber sensor to various fields in which the shear force is dominant with the integration of rubber and electrodes using our proposed fabrication process [1]. The unclear inclination of textures such as the surfaces of papers, cloth, and skin with make-up can be clarified using the MCF rubber sensor. The novelty of the present report is the viability of sensing the shear force and the softness of the object by stretchable and elastic rubber made of single material.

## 2. Electrical Properties for Normal and Shear Motions

Using the percolation technique of a permeating agent into the MCF rubber [1] and the adhesion technique of electrode wires on the MCF rubber [2], we proposed a consummate fabrication of an MCF rubber sensor [1]. The outer Q rubber shell of the MCF rubber sensor had the dimensions 16.5 mm × 24 mm in size and 4.5 mm thick, and the inner Q rubber was 16 mm × 21.5 mm and 4 mm thick. First, we investigated the electrical properties of the MCF rubber sensor produced using the consummate fabrication process under normal or shear forces.

### 2.1. Compression

From the aspect of piezo-elements, the electric characteristics are predominantly categorized into piezo-electricity and piezo-resistivity [17]. The former is evaluated as built-in electricity (i.e., built-in voltage and current), which is caused by the ionized molecules and particles of the rubber latex, oleic acid-coated around the Fe_3_O_4_ of the MF, water, and PVA. They functionalize an acceptor-like p-type semiconductor (corresponding to A^−^) and a donor-like n-type semiconductor (corresponding to D^+^) [18,19]. However, the latter requires the application of voltage through a power supply so that resistivity changes, whose mechanism can be explained by typical tunnel theory [18]. 

We used a previously proposed normal force experiment (NFE) apparatus [20] and it is shown in Figure A1, in Appendix A. We measured the electrical properties of the MCF rubber sensor under compression. In the NFE, the upper electrode was moved to be in contact with the MCF rubber sensor, which was sandwiched between the electrodes, onto the lower electrode using an actuator at a pressing speed of 10 mm/min. The pressing normal force was measured using a load cell installed in the actuator. The actuator used a commercial, small tensile testing machine (SL-6002, IMADA-SS Co. Ltd., Toyohashi, Japan). The paired electrodes used had the same 7 mm × 7 mm square form. For piezo-electricity, we measured the induced voltage from the electric wires of the MCF rubber sensor. For piezo-resistivity, by the application of a voltage to the MCF rubber, we measured the electrical current passing through the MCF rubber sensor at compression. This was accomplished using an electric power supply of 10 V and electric resistance of 1.8 kΩ in the electric circuit for piezo-resistivity.

The MCF rubber sensor was produced using the proposed consummate fabrication process [1]. “Adhesive MCF rubber 2” consisted of 0.5 g of Na_2_WO_4_·2H_2_O, 3 g of water, 0.75 g of MF (W40, Ichinen-Chemicals Co., Ltd., Shibaura, Japan), 3 g of KE1300T (Shin-Etsu Chemical Co. Ltd., Tokyo, Japan), 3 g of PVA, 3 g of NR-latex (Ulacol, Rejitex Co. Ltd., Atsugi, Japan), 3 g of CR-latex (671A, Showa Denko Co. Ltd., Tokyo, Japan), 3 g of TiO_2_ (Anatase type, Fujifilm Wako Pure Chemical Co., Ltd., Osaka, Japan), and 3 g of Ni. “Porous MCF rubber 1” consisted of 0.5 g of Na_2_WO_4_·2H_2_O, 3 g of water, 0.75 g of MF, 3 g of KE1300T, 3 g of NR-latex (Ulacol), 3 g of CR-latex (671A), 3 g of TiO_2_, and 3 g of Ni. It was permeated with glycerin. “MCF rubber 3” consisted of 0.75 g of MF, 3 g of NR-latex (Ulacol), 3 g of CR-latex (671A), 3 g of TiO2, and 3 g of Ni. Here, we used KE1300T instead of the KF96 in the figure of the proposed consummate fabrication process shown in [1]. The MCF rubber liquid was electrolytically polymerized, whereby a static magnetic field of 312 mT was applied to a pair of two stainless electrodes with a 1-mm gap using permanent magnets as paired opposites via the application of a constant electric field at 2.7 A. The Ni powder had particles in the order of microns and bumps on the surface (No. 123 by Yamaishi Co. Ltd., Noda, Japan). The MF used was 40 wt% Fe_3_O_4_. The MCF rubber sensor produced in this study was the same as that in the previous one [1]. Figure 1a shows the induced voltage as piezo-electricity in response to the repeating pressure and Figure 1b the electric current as piezo-resistivity in response to a pressure. For the former, as the pressure increased, the induced voltage increased and subsequently decreased to zero. This tendency can be explained by the fact that, as the counter negative and active ions become closer, the induced voltage increases. As the pressure increases, the counter ions come into contact and this phenomenon has previously been clarified [17]. In contrast, for the latter, because the outer Q rubber acted as a buffer to the applied pressure, the electric current reacted slightly later than the pressure, which was a peculiar property. Regardless of the Q rubber buffer, the MCF rubber sensor is attractive to the normal force.

### 2.2. Shear Motion

We used a previously used same shear force experiment (SFE) apparatus [20] to measure the electric property of the MCF rubber sensor under shear motion, as shown in Figure A2 in the Appendix B. The power supply was 10 V and the electric resistance 1.8 kΩ in the electric circuit for piezo-resistivity. By applying a voltage, we measured the electric current in the MCF rubber sensor, which was the same as that in NFE. The MCF rubber sensor 4 was placed under an acrylic resin body 2 on which two electrodes of the stainless plate were attached. As Figure 2 shows, the MCF rubber sensor 4 came into contact with a rubbed object 7 and was moved parallel to the object by adjusting a constant height interval by adjustment device 1 along the object surface using actuator 6 at a sweeping distance of 50 mm. Our used range of sweeping velocity was less than 4 mm/s. The pressing normal force was measured using a load cell that is embedded inner the base on which the rubbed object 7 is settled. Our used range of initial normal force means the normal force before the sensor’s movement was less than 5 N. The sweeping distance was measured using laser displacement 5. A hard, non-electric body with φ 0.5 mm was interposed between the MCF rubber sensor 4 and the acrylic resin body 2 so that the MCF rubber sensor could be contacted correctly [20]. This contact method was previously used [20] to be more effective when the contact area was smaller than the MCF rubber as a whole, and the MCF rubber is very slightly bent owing to the bending effect of the MCF rubber. The shear force was measured using a load cell 3 attached to the actuator 6.

Initially, the rubbed object was sandpaper that was attached to a flat plate with a rough surface. The sandpaper was prepared with various values of surface roughness and was optimal for investigating the effect of surface roughness on the MCF rubber sensor. The electric current of the MCF rubber sensor is shown in Figure 3. The MCF rubber sensor moved from a positive location to a negative one (from right to left). Using a surface roughness measuring device (SJ-400, Mitutoyo, Co. Ltd., Kawasaki, Japan), we observed that #40 sandpaper had 56.2 μm *Ra*, 216.7 μm *Ry*, and 68.2 μm *Rq*; #180 sandpaper had 14.1 μm *Ra*, 74.4 μm *Ry*, and 17.2 μm *Rq*; #280 sandpaper had 9.75 μm *Ra*, 55.8 μm *Ry*, and 12.6 μm *Rq*; and #1000 sandpaper had 3.32 μm *Ra*, 21.6 μm *Ry*, and 4.1 μm *Rq*. When the roughness curve represents *z*(*x*), the arithmetic average height *Ra* and root mean square height (*Rq*) are shown in Equation (1) at a reference length (*L*). *Ry* is the maximum height.
(1)$$Ra=\frac{1}{L}\int_{0}^{L}\left|z\left(x\right)\right|dx,\;\;Rq=\sqrt{\frac{1}{L}\int_{0}^{L}z^{2}\left(x\right)dx}\;$$

The electric current changes by three parameters: the surface roughness of the rubbed object, sweeping velocity, and normal force. Even with an outer shell of a Q rubber coating around the MCF rubber sensor [1], the MCF rubber sensor was sensitive to both shear and normal forces.

To ensure that the obtained lines in Figure 3 correctly represented shear force, we adopted #180 sandpaper with a 2.1-mm/s sweeping velocity and 3.7-N normal force, and then compared the electric current with the shear force measured using a load cell (Figure 4a). By considering that the electric pole of the electric current was reversed to that of the shear force, a quantitative changing tendency was coincident to both of them. Therefore, the electric current characteristics on the shear force were reactions to the convex and concave shapes of the object. Subsequently, the friction coefficient calculated by dividing the measured shear by the normal forces characterized the surface roughness of the object (Figure 4b). The change in the friction coefficient was non-linear overall, although a constant existed depending on the experimental conditions, such as in for #40, #280, and #1000 sandpapers. The non-linearity was due to the repeated deformation and restoration of the multi-layered MCF rubber sensor by the force created between the layers under shear motion (Figure 4c).

In the following paragraphs, we investigate the electric current affected by three parameters: the surface roughness of the rubbed object, sweeping velocity, and normal force. From the observation that the electric current corresponds to the shear force (Figure 4), the electric current can be evaluated using the same ordinary arithmetic values on the roughness curve measured using ordinary surface-roughness-measuring devices such as SJ-400: *Ra*, *Rq*, and *Ry*. Therefore, with respect to the electric current, *Ra_,E_*, *Rq_,E_*, and *R_y,E_*, which corresponds to *Ra*, *Rq*, and *Ry*, are calculated. These are affected by the surface roughness of the object (*RMI_Ra*, *RMI_Rq*, and *RMI_Ry*), sweeping velocity (*v*), initial normal force (*N_o_*) (which is the normal force before sweeping), as well as by *Ra_,N_*, *Rq_,N_*, and *Ry_,N_*, which are evaluated on the normal force and correspond to *Ra*, *Rq*, and *Ry*, and by *Ra_,S_*, *Rq_,S_*, and *Ry_,S_*, which are evaluated on the shear force and correspond to *Ra*, *Rq*, and *Ry*.

The relationships among these parameters are shown in Figure 5, Figure 6, Figure 7, Figure 8, Figure 9 and Figure 10. To investigate the effect of the parameters on the electric current of the MCF rubber sensor, we adopt the coefficient of determination (*R*^2^), which can exhibit linear regression and is also shown in the figures. We adopt the linear regression because of the convenience that if these parameters have a linear formula, when one of these parameters is known beforehand, all of the other parameters can become clear. The arrangement of the linear relationship is significant in the engineering development by using our sensor. In general, we can conclude that the correlation is strong when *R^2^* is more than 0.7 and is weak when *R*^2^ is less than 0.4 and greater than 0.2. When *R*^2^ is small, the formula might be arranged nonlinear relationship. As Figure 5 shows, the electric current has a strong correlation with the surface roughness and their correlations on *Ra* and *Rq* are stronger than on *Ry*. As Figure 6 and Figure 7 show, the electric current has a moderate correlation with the normal force and a stronger correlation with the normal force than *N_o_*. As Figure 7 and Figure 8 show, the electric current has a stronger correlation with the normal force than with the shear force and has stronger correlations with normal and shear forces on *Ra* than on *Rq* and *Ry*. As Figure 9 shows, the electric current has a weak correlation with *v*. However, Figure 10 shows that the shear force has a strong correlation with the sweeping velocity, which means that the dynamic frictional force is a function of the sweeping velocity that changes non-linearly. This nonlinearity is indicated in Figure 4c.

As a result, the MCF rubber sensor is sensitive to both normal and shear forces, regardless of the Q rubber buffer coated around the fabricated MCF rubber. Therefore, we expect that it can be used for the haptic sensing of diverse objects by either normal or shear motions. The feasibility of this method is discussed in the following section.

Incidentally, the cross sensitivity is also significant because it is relevant to the sensor’s accuracy. As for the present MCF rubber sensor, the data involving the cross sensitivity are obtained by measurement. The cause is that the particles and molecules of the MCF rubber are mixed and that the electric property generates inner the hybrid particles and molecules notwithstanding the existence of magnetic clusters aligned along the single direction, such as a needle in which electricity transmits. Therefore, the cross sensitivity cannot be divided from the obtained data.

## 3. Electrical Properties of Diverse Object

The condition of rubbing an object creates several opportunities for haptic sensing. We investigated the feasibility of the MCF rubber sensor on diverse objects under normal or shear motions.

### 3.1. Soft Object

The softness of the rubbed object was adjusted using silicone gel. We first investigated its softness by using the MCF rubber sensor through compression and then investigated its characteristics under shear motion.

For compression, we attached the MCF rubber sensor 3 under a rigid acrylic resin body 1 in contact with the silicone gel 4 which is settled on the base 2 (Figure 11). The silicone gel 4 was created from KE1300T and a mixing thinner at various weight-ratios of the thinner to KE1300T: 0.5, 1.5, 2.0, 2.5, 3.5, 4.0, and 5.0. By comparing the softness of the silicone gel, we also used a rigid body of iron. Using the NFE described previously, a rigid acrylic resin body was installed on the actuator. By applying pressure, the electric current in the MCF rubber sensor 3, as piezo-resistivity, was measured (Figure 12a). Because of the elasticity of the silicone gel, the electric current fluctuated slightly. Therefore, we investigated the elasticity of the silicone gel based on the relationship between pressure and compressive shear rate (Figure 12b). The results of the iron body corresponded to those in Figure 1b. As the enhancement of the amount of thinner increased, the elasticity weakened. From Figure 12b, we can posit that the relationship of the pressure to the shear rate has an approximate quadratic function. The coefficient of the secondary term of the approximated quadratic function is dominant such that it might represent the stiffness of the silicone gel, which is indicated by the parameter *a* in this paper. The electric current, *Ra_,E_*, *Rq_,E_*, and *Ry_,E_*, as in the previous section, are shown with the parameter *a* in Figure 12c. “*Max*” in the figure is the maximum value of the electric current at compression. *Ra_,E_*, *Rq_,E_*, *Ry_,E_*, and *Max* are linear to *a*, which means that the electric current can be presented by the stiffness of the silicone gel.

For shear motion, the MCF rubber sensor was rubbed along the surface of the silicone gel using the SFE instrument and measuring method described in the previous section. No conventional instrument for measuring shear force or friction coefficient without destroying the gel exists. The MCF rubber sensor did not break the gel. The electric current of the MCF rubber sensor as piezo-resistivity is shown in Figure 13 under shear motion. The MCF rubber sensor moved from a positive location to a negative one (from right to left). The electric current included the effect of normal force and sweeping velocity, as discussed in the previous section. However, in this study, both the initial normal force and sweeping velocity were maintained at a constant. The deforming motion of the silicone gel is illustrated in Figure 14a. At the beginning of the rubbing, the MCF rubber sensor slid smoothly on the silicone gel, and then the silicone gel gathered in billows (a and b in Figure 14a). Subsequently, the silicone gel returned to its original flat shape, and the gathered silicone gel passed under the MCF rubber sensor (c and d in Figure 14a). This deformation was repeatable. Therefore, the friction coefficient had several peaks or fluctuations (Figure 14b). The electric current can be used to demonstrate the friction coefficient.

In contrast, when the softness of a soft object such as the silicone gel is measured, it is pushed and its elasticity measured, as discussed in the previous section using the NFE. In general, the relationship between normal and shear motions has not been conducted on a soft object before. Therefore, we investigated the relationship between the elasticity and the friction coefficient of the silicone gel. The former is denoted by the parameter *a* and the latter by the electric current of the MCF rubber sensor. *Ra_,E_*, *Rq_,E_*, and *Ry_,E_* denote the friction coefficients and thus the elasticity and friction coefficient are related (Figure 14c). The indexes a, b, c, and d in Figure 14c correspond to a, b, c, and d in Figure 14a respectively. *Ra_,E_*, *Rq_,E_*, and *Ry_,E_* exhibit two predominant peaks, which correspond to b and d in Figure 14a. 

Therefore, the normal and shear motion of the MCF rubber sensor on the soft body can be used for the haptic sensing of any soft body.

### 3.2. Paper

The parameters of sensing a soft body are softness and texture. The former can be measured by applying a normal force and elucidated from the results obtained in the previous section. In contrast, the latter is measured by applying shear force and has not been clarified yet except using silicone gel in the previous section. First, because paper is a typical texture of a soft body, it is discussed here. The MCF rubber sensor was rubbed along the surface of a paper using the SFE instrument and measured using a method described in the previous section. Figure 15 shows images of various forms of paper, labeled A–D. The electric current of the MCF rubber sensor with respect to piezo-resistivity is shown in Figure 16a. The MCF rubber sensor was rubbed along the surface of the paper using the same SFE instrument and measuring method described in the previous section. For comparison, the measurement of the acrylic resin is also shown. The *Ra_,E_*, *Rq_,E_*, and *Ry_,E_* obtained from the electric current of Figure 16a, and the *RMI_Ra*, *RMI_Rq*, and *RMI_Ry* of the paper measured using the ordinary surface-roughness-measuring device are shown in Figure 16b. In contrast to the *RMI_Ra*, *RMI_Rq*, and *RMI_Ry* of acrylic resin, which were sufficiently low to be smooth, *Ra_,E_*, *Rq_,E_*, and *Ry_,E_* were larger because of the peak or fluctuation created by the surface roughness owing to the layered fabrication of the MCF rubber sensor (Figure 4 and Figure 14). In contrast, the surface of A was coarser than B, C, and D; therefore, its *RMI_Ry* was larger. However, for B, C, and D, *Ra_,E_*, *Rq_,E_*, and *Ry_,E_* were larger than *RMI_Ra*, *RMI_Rq*, and *RMI_Ry*. In particular, the *Ry_,E_* of B was the largest because the fiber of the newspaper was coarse. These results can be guessed from the images by scanning electron microscope (SEM) that fibers of A and B are large. They indicate that the fineness of the paper cannot be subtly measured by sweeping the diamond needle of an ordinary surface-roughness-measurement device such as SJ-400.

### 3.3. Cloth

Cloth is a typical fine and soft material. Figure 17 shows images of various cloths, labeled A–H. The electric current of the MCF rubber sensor as piezo-resistivity is shown in Figure 18a. The MCF rubber sensor was rubbed along the surface of the cloth using the same SFE instrument and measuring method described in the previous section. Figure 2 corresponds to Figure 17b. Figure 18b shows the comparison between *Ra_,E_*, *Rq_,E_*, and *Ry_,E_* obtained from the electric current in Figure 18a and *RMI_Ra*, *RMI_Rq*, and *RMI_Ry* of paper measured using ordinary surface-roughness measurement. *RMI_Ra*, *RMI_Rq*, and *RMI_Ry* were observed to be inaccurate values that the surface roughness of the cloths cannot be measured by sweeping the diamond needle of ordinary surface-roughness-measurement devices such as SJ-400. Therefore, the cloths were torn by SJ-400, particularly the large fibers of cloth B (Figure 17f). However, the MCF rubber sensor can measure the surface roughness without breaking the surface, and the *Ra_,E_*, *Rq_,E_*, and *Ry_,E_* of the different cloths could be obtained. In particular, the surface roughness of cloth B, fabricated using many large and soft fibers, could be obtained. Regarding the denseness and alignment of fibers, the more neatly the fibers align, such as for H, the larger *Ry_,E_* becomes. However, even if the fibers form neat lines, *Ry_,E_* changes significantly or slightly depending on whether the cloth is soft or hard, such as cloths E and F. Therefore, we investigated from the perspective of the material fibers, A and B had fibers made of cotton. The fiber’s shape is slightly twisted or wrung one as shown in Figure 17i,j. C, D, and G had both chemical fibers made of polyester and wool. The fiber’s shape is smooth one without twisting as shown in Figure 17k,l,w. E, F, and H had fibers made of wool. The fiber has small jagged cuticles as shown in Figure 17u,v,x. The *Ry_,E_* of the cloths made of cotton, such as A and B, and those of wool such as E and H, tended to be small. The *Ry_,E_* of F was larger than that of E and H because the lump of fibers of F was larger than that of E and H. At the cloth including chemical fibers such as C and D, *Ry_,E_* tended to become larger. The *Ry_,E_* of G was smaller than that of C and D because its fibers were smaller than that of C and D.

### 3.4. Plants

Subsequently, we investigated a comparatively rough-surface, convexo-concave body. We attempted the test on the back surfaces of hydrangea and cherry leaves (Figure 19). The MCF rubber sensor was rubbed along the surface of the leaf using the same SFE instrument and measuring method described in the previous section. By crossing the leaf veins as shown in Figure 19b,d, the MCF rubber sensor captures their convexo-concave shapes using the electric current of the MCF rubber sensor as piezo-resistivity (Figure 19a,c).

### 3.5. Large Convexo-Oconcave Body

Additionally, we investigated the surface roughness of a large convexo-concave body, which was a metal body fabricated from brass (Figure 20). The height of the convexo-concave body in Figure 20a was significantly larger than those of Figure 19b,d. The MCF rubber sensor was rubbed along the surface of the body using the same SFE instrument and measuring method described in the previous section. By crossing the body, the MCF rubber sensor captured its convexo-concave shape using the electric current of the MCF rubber sensor as piezo-resistivity (Figure 20b).

### 3.6. Cosmetic

Finally, we examined the surface roughness of human skin as shown in Figure 21a. The MCF rubber sensor was rubbed along the surface of the little finger using the same SFE instrument and measuring method described in the previous section. By crossing the little finger, the MCF rubber captured its convexo-concave shape using the electric current of the MCF rubber sensor as piezo-resistivity (Figure 21b). A large electric current occurred at the first and second joints and the base of the nail. 

In contrast, the surface roughness of the human skin coated with cosmetic cream was smoother than that without one. Therefore, if we apply the MCF rubber sensor to the human body, we can measure and evaluate the degree to which the skin is covered by cosmetics or the quality of hair with cuticles.

## 4. Temperature Sensing

The MCF rubber can be used for diverse sensing including the force sensing of temperature and a wide range of electromagnetic waves involving photovoltaics, irradiation, and infrared rays [4]. In this study, we investigated temperature sensing. We will present the investigations on photovoltaics and irradiation in subsequent reports [21].

As Figure 22a shows, we measured the electric current as the piezo-resistivity of the MCF rubber sensor immersed in water into which hot water was poured. The MCF rubber sensor and measurement method were the same as described in the previous section. We simultaneously measured the temperature adjacent to the MCF rubber sensor using a thermocouple. Figure 22b shows the change in the electric current as a function of water temperature. At the initiative period of less than approximately 2 s, the electric current was independent of the increase in water temperature because of the buffer of the outer shell of the Q rubber coat around the MCF rubber sensor. Subsequently, the electric current changed according to the increase in water temperature, which indicated a good agreement.

## 5. Conclusions

An MCF rubber sensor produced using the consummate fabrication process presented in the last report is sensitive to both normal and shear forces. Therefore, it can be used for haptic sensing of diverse objects using either normal or shear motions. In addition, the MCF rubber sensor is sensitive to softness and temperature. The unclear inclination of the texture of the surfaces of paper, cloth, skin applied with cosmetics, etc., can be clarified using the MCF rubber sensor. The novelty of the present report is the feasibility of sensing the shear force and the softness of the object by stretchable and elastic rubber made of a single material. Our proposed sensing technique could contribute to the enterprising development of their production.

Using shear motion, the electric current of the MCF rubber sensor reacts to the convex and concave shapes of the object. Subsequently, the friction coefficient can be measured using the MCF rubber sensor and it is non-linear because of the repeated deformation and restoration of the multi-layered MCF rubber sensor by the force created between the layers under shear motion. The electric current can be evaluated by the same ordinary arithmetic values such as *Ra_,E_*, *Rq_,E_*, and *Ry_,E_* on its roughness curve measured using an ordinary surface-roughness-measuring device: *Ra*, *Rq*, and *Ry*. *Ra_,E_*, *Rq_,E_*, and *Ry_,E_* are affected by the surface roughness of the object, sweeping velocity, and normal and shear forces. Because the outer dimensions of the MCF rubber sensor are reasonable (16.5 mm × 24 mm × 4.5 mm), the MCF rubber sensor can be installed on a fingerstall, is portable, and is useful in measuring the surface roughness of any automobile, airplane, and large ship that cannot be broken for measurement using ordinary surface-roughness-measuring devices. The MCF rubber sensor is a novel surface-roughness-measuring device.

For engineering applications using the MCF rubber sensor, we investigated the feasibility of the MCF rubber sensor for the haptic sensing of diverse surfaces under normal or shear motions.

First, for a soft body, using silicone gel, the electric current of the MCF rubber sensor can represent the softness or elasticity of the silicone gel under compression. Moreover, the electric current of the MCF rubber sensor under shear motion can indicate the friction coefficient of the silicone gel. The friction coefficient and elasticity are related; the friction coefficient to the elasticity has two peaks. In general, the relationship between normal and shear motions has not yet been investigated for soft objects. However, we can investigate the relationship between the elasticity and the friction coefficient of a soft object.

Second, the fineness of paper texture can be measured using the MCF rubber sensor under shear motion. Additionally, the sensor can be used to measure the surface roughness of a cloth as a typical fine and soft object, which an ordinary surface-roughness-measurement device cannot do. Thus, the MCF rubber sensor is applicable to the fields of paper and fashion or apparel manufacturing industries. Until now, we determined the texture of paper and cloths by touching them with hands, through which the texture cannot be evaluated numerically. However, by using the MCF rubber sensor, their texture can be evaluated quantitatively by *Ra_,E_*, *Rq_,E_*, and *Ry_,E_*.

Third, the MCF rubber sensor can capture the shapes of convexo-concave bodies, such as the surface of plant leaves and convexo-concave metal. Significantly small surface roughness, such as for a plant leaf, can be evaluated using *Ra_,E_*, *Rq_,E_*, and *Ry_,E_*. However, large surface roughness, such as that of large convexo-concave metal bodies, can be measured directly.

Finally, the MCF rubber sensor can capture the convexo-concave shape of the human body and the surface roughness of skin to which cosmetics are applied. Therefore, if we apply the MCF rubber sensor to the human body, we can measure and evaluate the degree to which the skin is covered by cosmetics or the quality of hair with cuticles. Thus, the MCF rubber sensor is applicable to the cosmetics industry.

The MCF rubber sensor was observed to be effective in diverse haptic and temperature sensing. We expect that the MCF rubber applies to the force and temperature sensing for a wide range of electromagnetic waves involving photovoltaics, irradiation, and infrared rays. These investigations will be presented in subsequent reports. Thus, the MCF rubber sensor can be applied to a wide range of engineering fields. Furthermore, we can develop our proposing MCF rubber sensor with using the Artificial Intelligent (AI) model. The possibility that the sensing gets closer to the virtual sensing of the human skin might be facilitated. The technique would be expectable enough to be useful in the field of virtual sensing of skin installed in human being and robot.

## Figures and Tables

**Figure 1 sensors-20-04674-f001:**
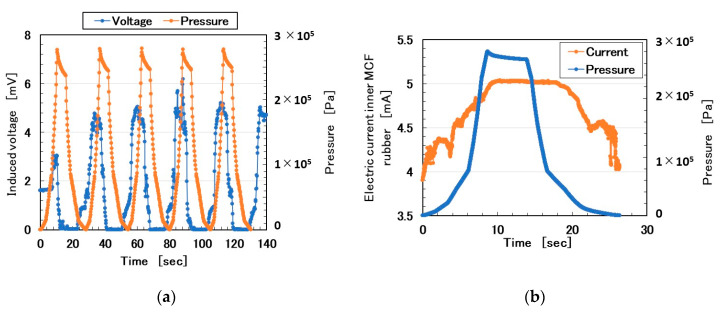
Electrical property of magnetic compound fluid (MCF) rubber sensor in response to repeating pressure: (**a**) for piezo-electricity; (**b**) for piezo-resistivity.

**Figure 2 sensors-20-04674-f002:**
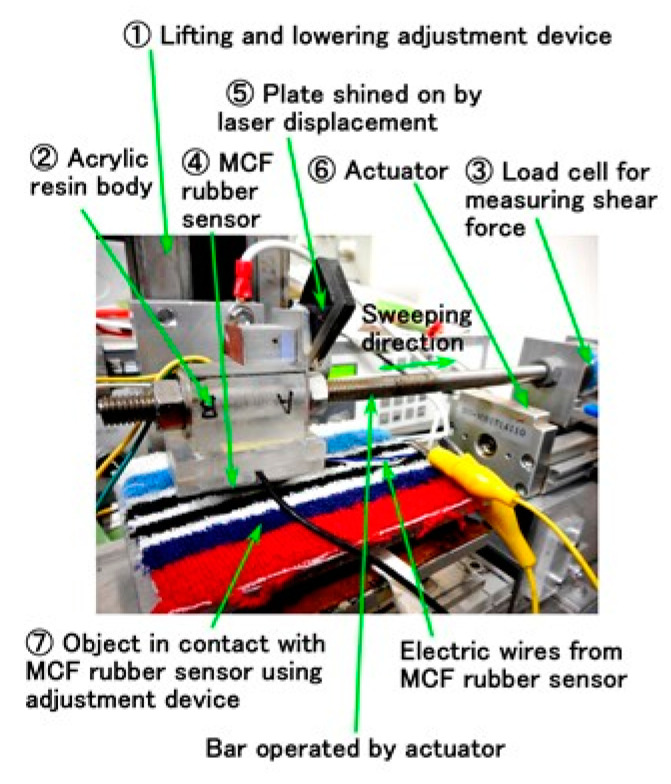
Image of experimental apparatus used to investigate the electrical property in the case of the shear force experiment (SFE).

**Figure 3 sensors-20-04674-f003:**
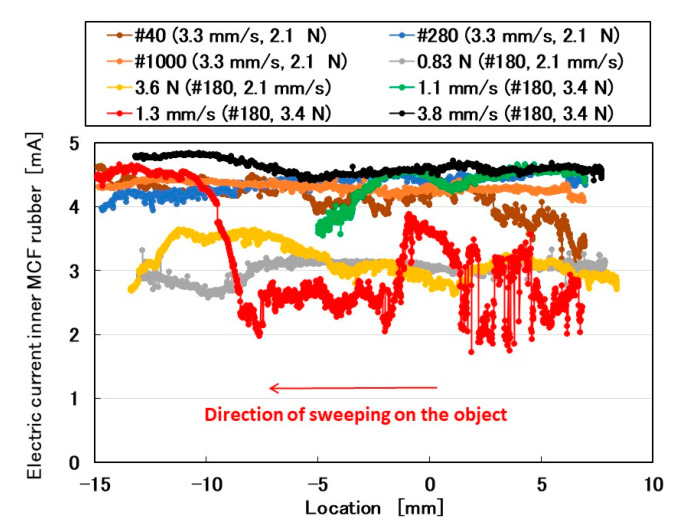
Electrical properties of MCF rubber sensor rubbing over sandpaper in the SFE: the force is presented as the initial normal force just before sweeping. The sweeping velocity is also presented.

**Figure 4 sensors-20-04674-f004:**
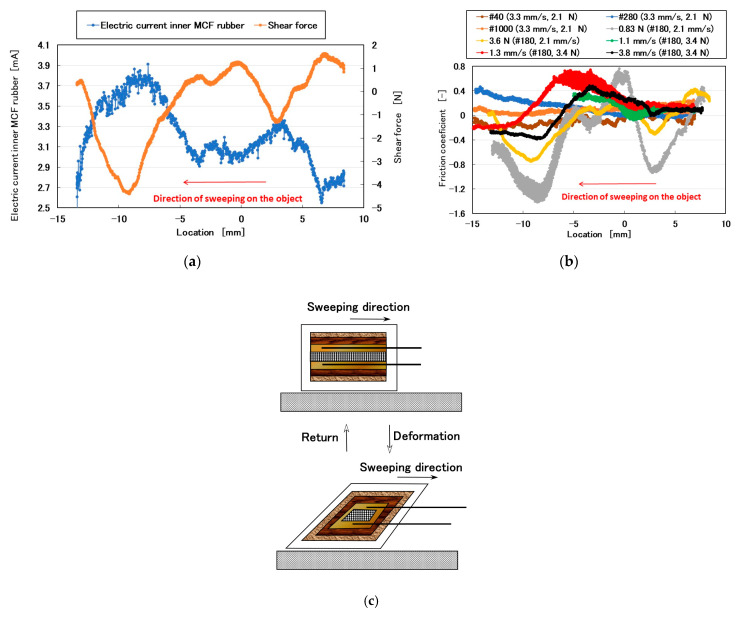
Characteristics of MCF rubber sensor under shear motion; (**a**) corresponding electric current to shear force; (**b**) friction coefficient; (**c**) model of MCF rubber sensor under shear motion.

**Figure 5 sensors-20-04674-f005:**
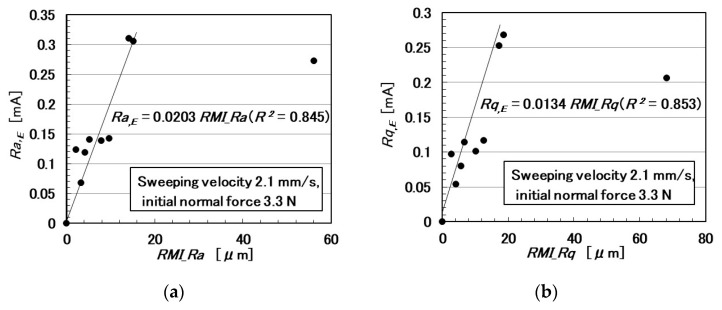
Relationship between electric current and surface roughness: (**a**) *Ra*.; (**b**) *Rq*.; and (**c**) *Ry*.

**Figure 6 sensors-20-04674-f006:**
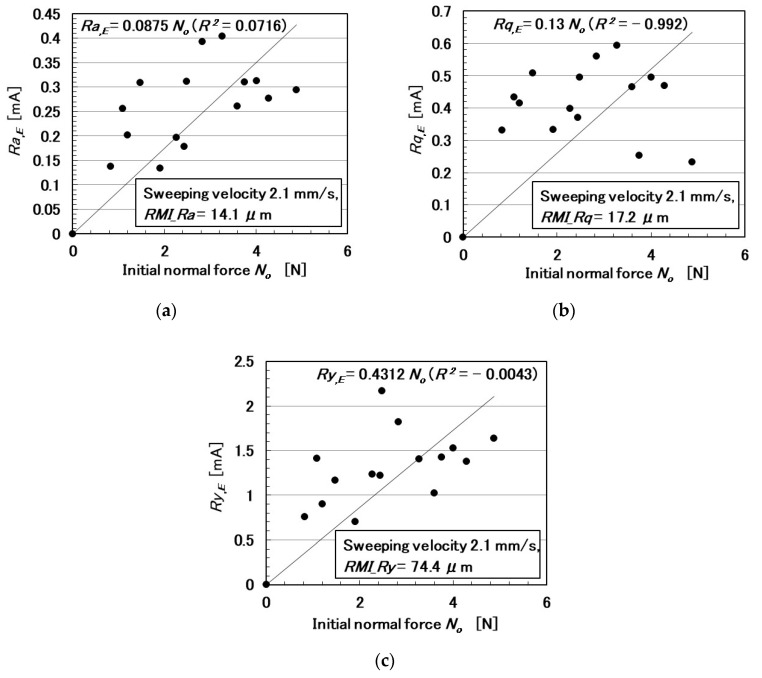
Relationship between electric current and initial normal force: (**a**) *Ra*.; (**b**) *Rq*.; and (**c**) *Ry*.

**Figure 7 sensors-20-04674-f007:**
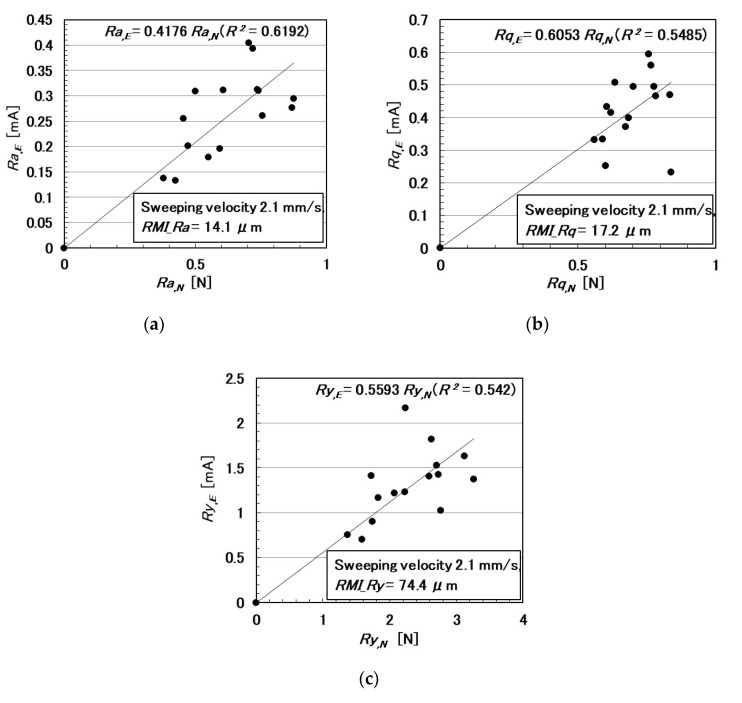
Relationship between electric current and normal force: (**a**) *Ra*.; (**b**) *Rq*.; and (**c**) *Ry*.

**Figure 8 sensors-20-04674-f008:**
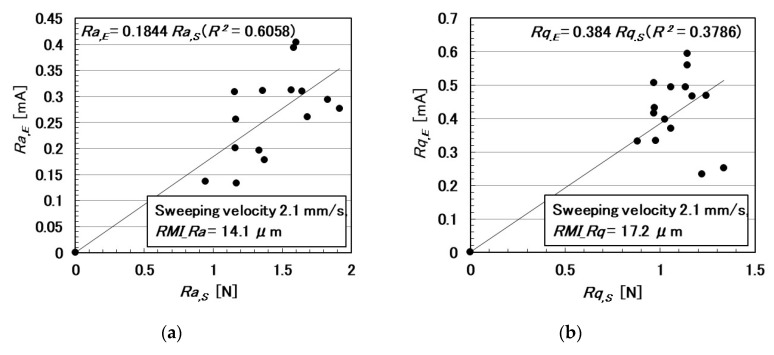
Relationship between electric current and shear force: (**a**) *Ra*.; (**b**) *Rq*.; and (**c**) *Ry*.

**Figure 9 sensors-20-04674-f009:**
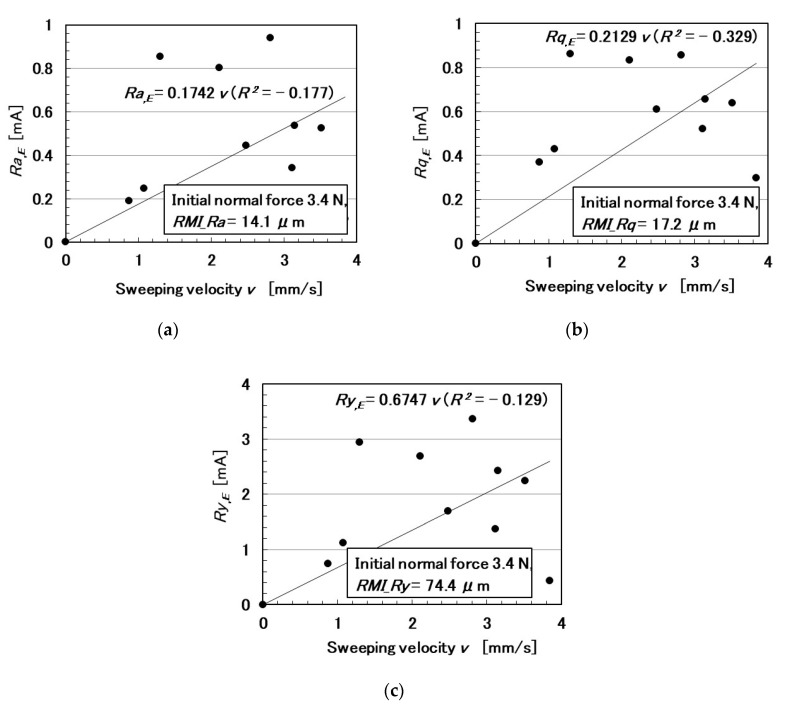
Relationship between electric current and sweeping velocity: (**a**) *Ra*.; (**b**) *Rq*.; and (**c**) *Ry*.

**Figure 10 sensors-20-04674-f010:**
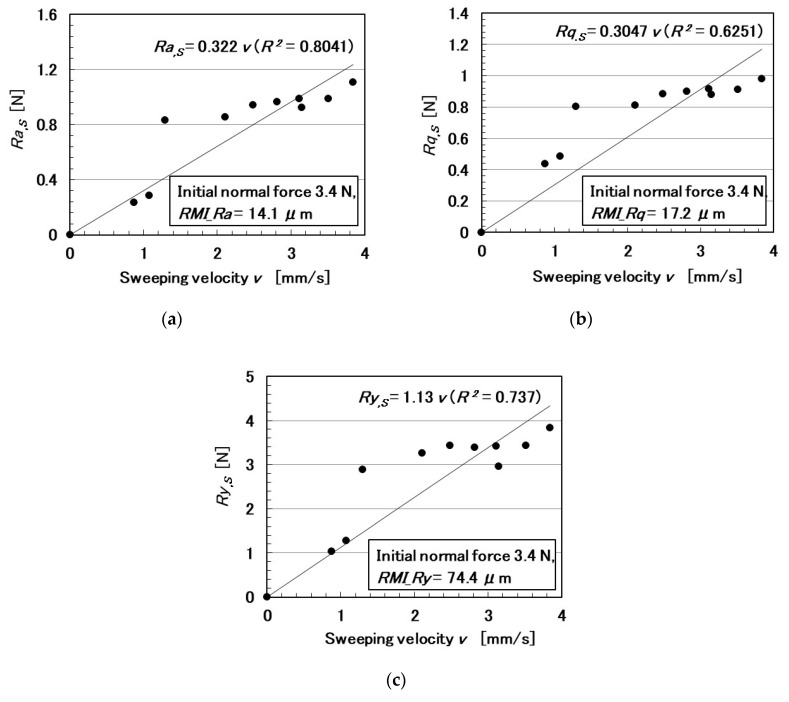
Relationship between shear force and sweeping velocity: (**a**) *Ra*.; (**b**) *Rq*.; and (**c**) *Ry*.

**Figure 11 sensors-20-04674-f011:**
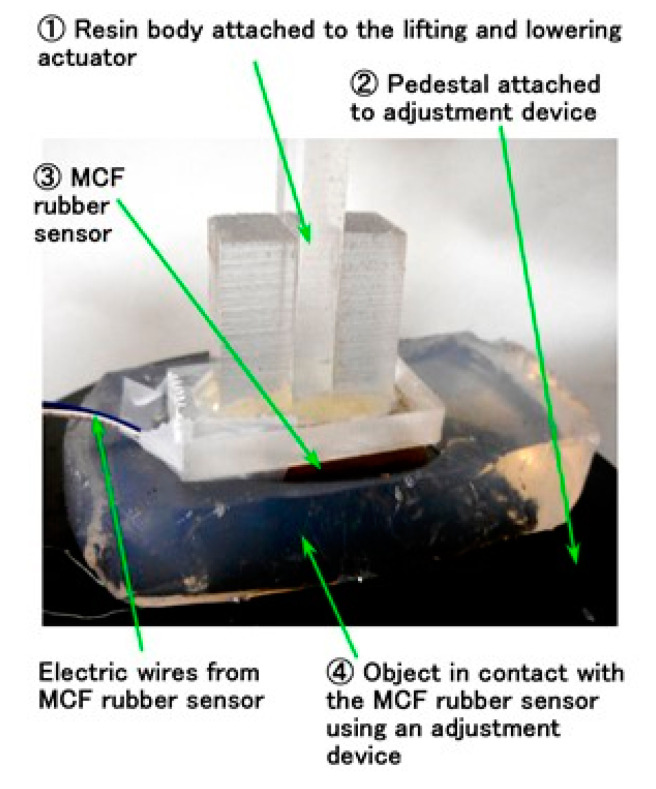
Image of touching the silicone gel by the MCF rubber sensor.

**Figure 12 sensors-20-04674-f012:**
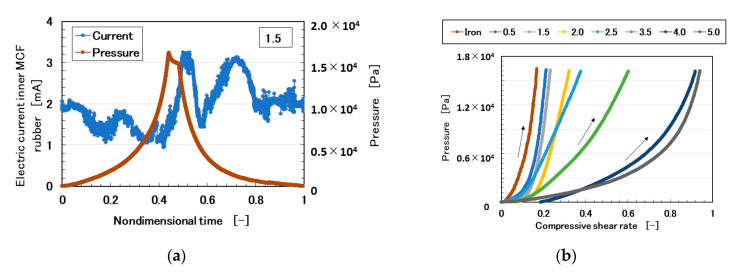
Results of the silicone gel in contact with the MCF rubber sensor under compression: (**a**) electric current and pressure with a ratio of involving thinner to KE1300T at 1.5; (**b**) relationship between pressure and compressive shear rate at various ratios of the thinner to KE1300T; (**c**) relationship of *Ra_,E_*, *Rq_,E_*, *Ry_,E_*, and *Max* to the parameter presented stiffness *a*.

**Figure 13 sensors-20-04674-f013:**
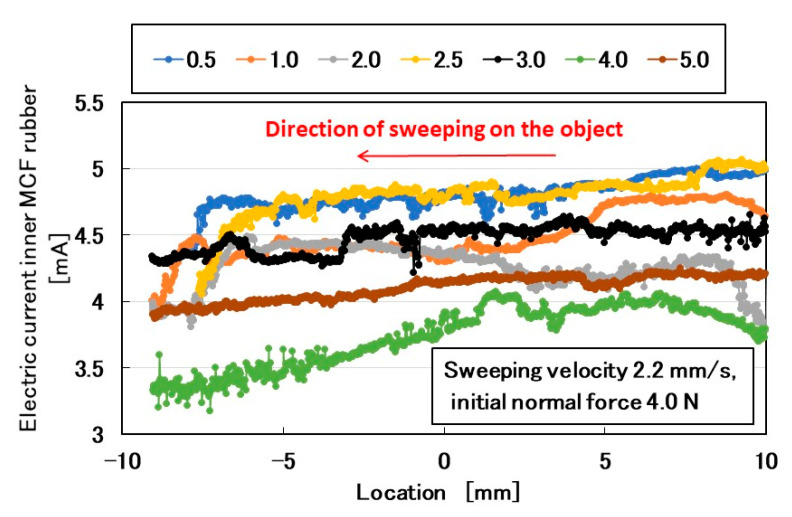
Electrical current of the MCF rubber sensor by rubbing silicone gel under shear motion at various ratio of the thinner to KE1300T.

**Figure 14 sensors-20-04674-f014:**
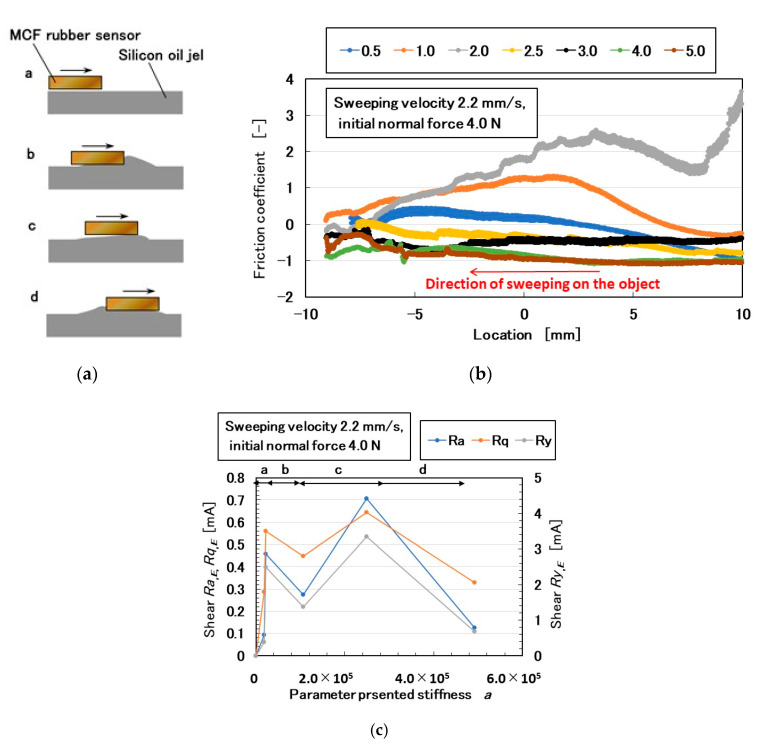
Model of results by touching the silicone gel with the MCF rubber sensor under shear motion: (**a**) schematic model of silicone gel deformation under shear force; (**b**) friction coefficient at various ratios of the thinner to KE1300T; (**c**) relationship of *Ra_,E_*, *Rq_,E_*, and *Ry_,E_* to the parameter presented stiffness *a*.

**Figure 15 sensors-20-04674-f015:**
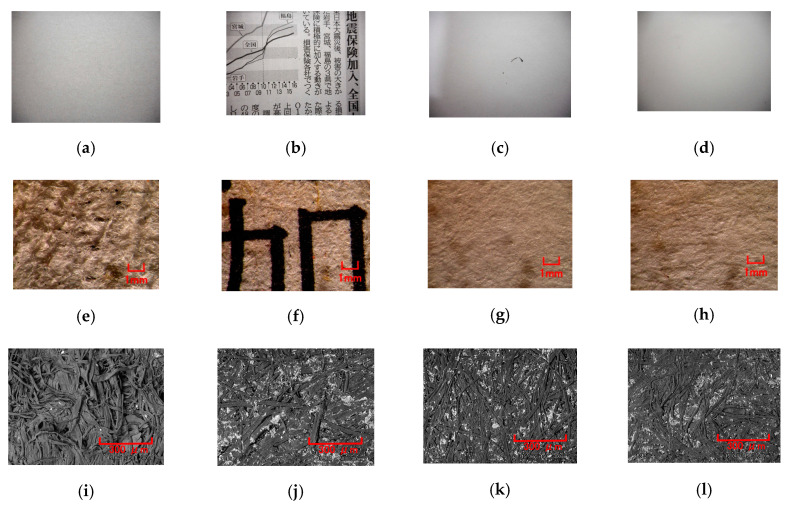
Images of various papers rubbed by the MCF rubber sensor: (**a**) toilet paper; (**b**) newspaper; (**c**) high-quality printing paper; (**d**) recycled printing paper; (**e**) and (**i**) microscopic image of A; (**f**) and (**j**) microscopic image of B; (**g**) and (**k**) microscopic image of C; (**h**) and (**l**) microscopic image of D; (**e**)–(**h**) by optical microscope, (**i**)–(**l**) by scanning electron microscope (SEM) with 150 magnification.

**Figure 16 sensors-20-04674-f016:**
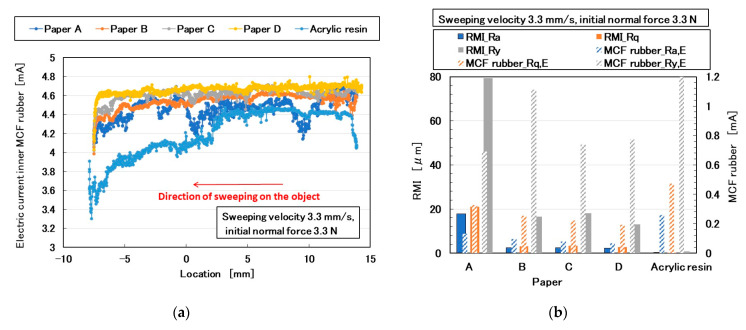
Results of paper by the MCF rubber sensor under shear motion: (**a**) electric current; (**b**) *Ra_,E_*, *Rq_,E_*, *Ry_,E_* of (**a**), *RMI_Ra*, *RMI_Rq*, and *RMI_Ry* of paper.

**Figure 17 sensors-20-04674-f017:**
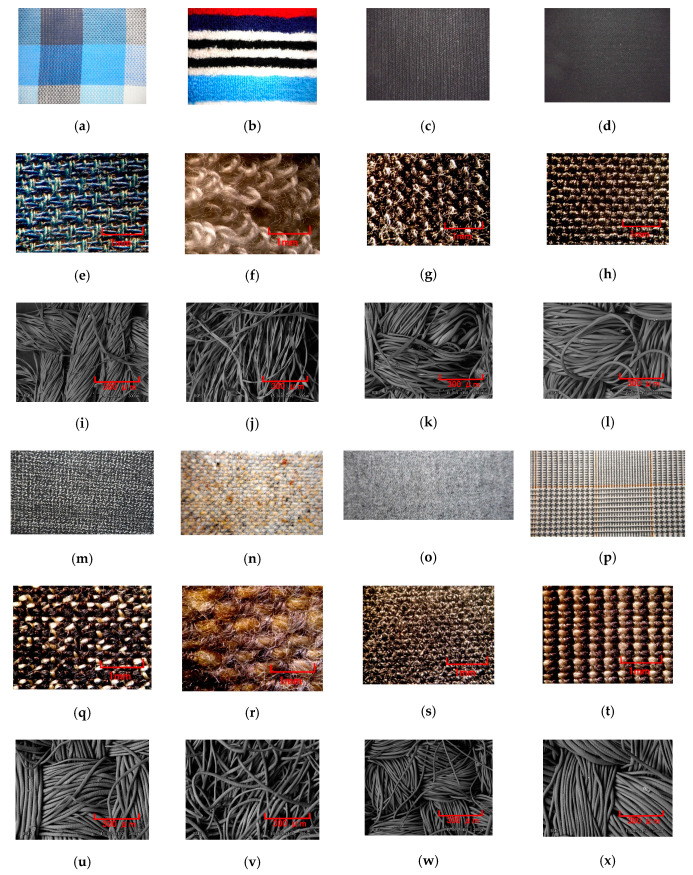
Images of various cloths rubbed by the MCF rubber sensor: (**a**) handkerchief; (**b**) washcloth; (**c**) cloth of trousers; (**d**) jacket for women; (**e**) and (**i**) microscopic image of A; (**f**) and (**j**) microscopic image of B; (**g**) and (**k**) microscopic image of C; (**h**) and (**l**) microscopic image of D; (**m**) E cloth of jacket for men; (**n**) F cloth of business suit for men in the winter; (**o**) G cloth of trousers for women; (**p**) H cloth of business suit for men in the summer; (**q**) and (**u**) microscopic image of E; (**r**) and (**v**) microscopic image of F; (**s**) and (**w**) microscopic image of G; (**t**) and (**x**) microscopic image of H; (**e**)–(**h**) and (**q**)–(**f**) by optical microscope; (**i**)–(**l**) and (**u**)–(**x**) by SEM with 150 magnification.

**Figure 18 sensors-20-04674-f018:**
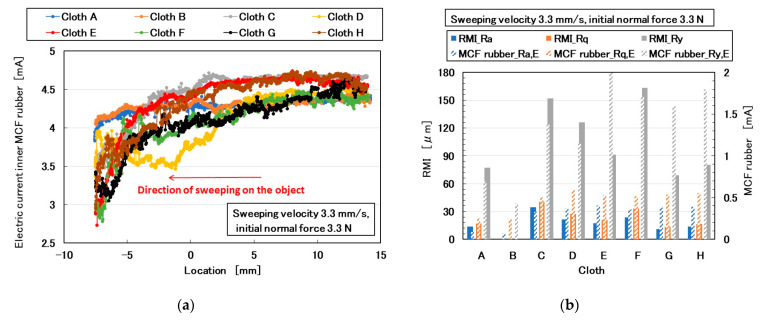
Results of cloth by the MCF rubber sensor under shear motion: (**a**) electric current; (**b**) *Ra_,E_*, *Rq_,E_*, *Ry_,E_* of (**a**), *RMI_Ra*, *RMI_Rq*, and *RMI_Ry* of cloth.

**Figure 19 sensors-20-04674-f019:**
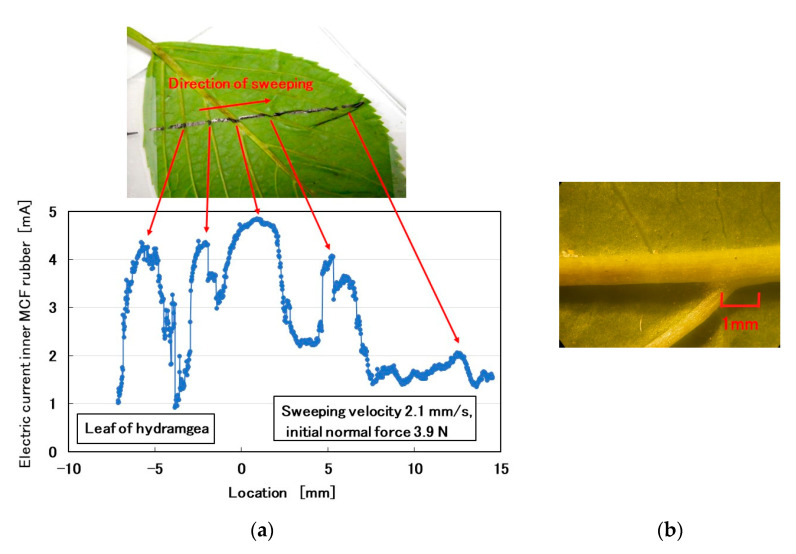
Rubbing the back of plant leaves using the MCF rubber sensor: (**a**) surface roughness of hydrangea leaf; (**b**) microscopic image of the hydrangea leaf; (**c**) surface roughness of cherry leaf; (**d**) microscopic image of the cherry leaf.

**Figure 20 sensors-20-04674-f020:**
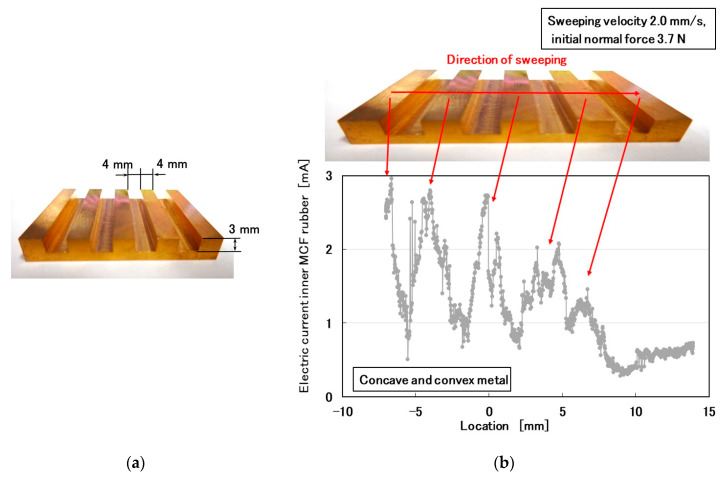
In the case of rubbing a large convexo-concave body by the MCF rubber sensor: (**a**) image of the body; (**b**) surface roughness of the body.

**Figure 21 sensors-20-04674-f021:**
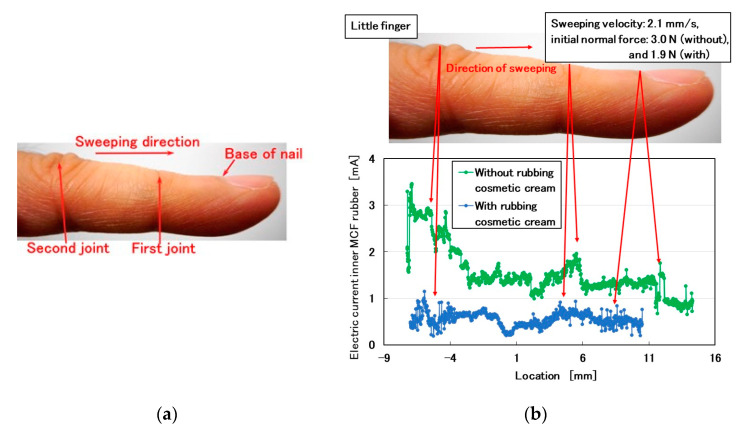
In case of rubbing little finger by the MCF rubber sensor: (**a**) image of the little finger and sweeping direction; (**b**) surface roughness of little finger coated with and without cosmetic cream.

**Figure 22 sensors-20-04674-f022:**
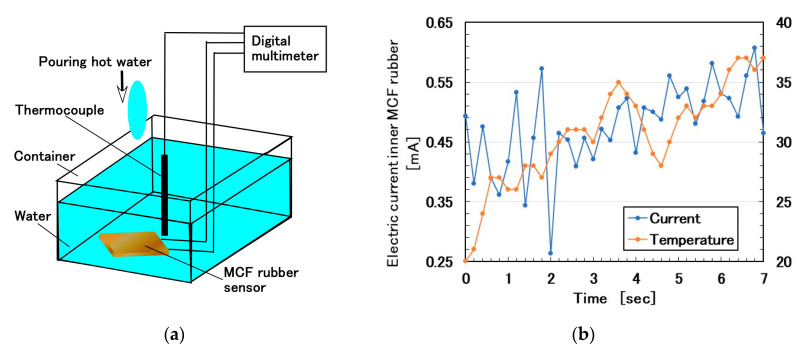
Temperature sensing of the MCF rubber sensor: (**a**) schematic diagram of experimental apparatus; (**b**) electric current of the MCF rubber sensor to the enhancement of temperature.

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
