# Peer review of "Enhancement of Diversity in Production and Applications Utilizing Electrolytically Polymerized Rubber Sensors with MCF: The Second Report on Various Engineering Applications"

_sensors, 2020, doi:10.3390/s20174674_

Round 1
Reviewer 1 Report
The article is devoted to a variety of practical application of the results obtained by the authors in the first part of the presented study. Research is well conducted, of broad interest, and has many technical applications. I recommend publishing of the article in its current form.
Author Response
Thank you for your valuable comments and suggestions for the present our report. According to your indication, we confirmed again to revise English by technical English editing in the overall document. Incidentally, they are numerous and minute enough not to be highlighted in the revised manuscript. I would appreciate for your kind cooperation.

Reviewer 2 Report
The paper under review presents Enhancement of Diversity on Production and Application Utilizing Electrolytically Polymerized Rubber Sensor with MCF. Overall the quality of the paper is good and it can be valuable for the sensor community. In my opinion, the article is suitable for publication in sensors.
I have the following minor suggestions:
- please revise English writing along the manuscript.
- Please emphasize the novelty of the current work. Also, more information should be given about previous work in the similar area.
Author Response
Thank you for your valuable comments and suggestions for the present our report. According to their comments, we would like to reply for them as follows.
Question 1: According to your indication, we confirmed to revise English by technical English editing in the overall document. Incidentally, they are numerous and minute enough not to be highlighted in the revised manuscript.
Question 2: According to your indication, the explanations of the novelty of the current work and of the information about previous works by comparing to our study are added in the Introduction, and Conclusion, which are for the effort to expand the range of studies considered and more clearly understood what the current work occupies.

Reviewer 3 Report
Thanks for the authors’ detailed work on MCF based sensor, which could have many engineering applications. It is recommended for publication after considering the following comments:
Line 84. The inner size dimension is “16 mm x 21.5 mm x 4.0 mm”? according to the following context.
Line 111-112, could you introduce particle size of TiO2 and Ni? And since there is water and Ni, did you observe any reaction?
Line 125 and Figure 1. Is there current response to “repeating” pressure?
Line 137, how to determine the resistance and do the resistance change during shear motion?
Figure 3 and Figure 4b, it looks like you have #180, why not do a 3.3 mm/s 2.1 N experiments, in that way with 4 points, it is more solid to tell the trend. In addition, it becomes so busy to understand using Figure 3. You can control the other two variables, only change one variable for one Figure to make it clearer to understand.
Figure 3, Figure 4a and 4b, Figure 13. Can you put a schematic picture as Figure 19 that clearly showing what the location means?
Figure 7, Figure 8. Is there any limitation for applying normal/shear force? Could you apply force between 0 and 1, or larger than 4?
For some Figures such as Figure 6 and 8. The linear relationship is not obvious, did you consider other relationship?
The MCF sensor you developed is flexible for many sensing applications. Do you try to do some experiments to investigate the cross sensitivity?
Try to improve the quality of some Figures like Figure 3, 4, 16. The original Figures in Excel should have higher quality.
Actually, the data you obtained are idea training data for artificial intelligence. You can develop AI models distinguish the true sensing data as well as extract real sensing results.
Author Response
Thank you for your valuable comments and suggestions for the present our report. According to their comments, we would like to reply for them as attachment.
